# Response of a Structure Isolated by a Coupled System Consisting of a QZS and FPS Under Horizontal Ground Excitation

**Richie Kevin Wouako Wouako** [1,*], **Sandra Céleste Tchato** [2], **Euloge Felix Kayo Pokam** [3],
**Blaise Pascal Gounou Pokam** [4], **André Michel Pouth Nkoma** [2,*], **Eliezer Manguelle Dicoum** [1]
and **Philippe Njandjock Nouck** [1]

[1]  Department of Physics, University of Yaoundé I, Yaoundé P.O. Box 812, Cameroon;
    madicoum@yahoo.fr (E.M.D.); pnnouck@yahoo.com (P.N.N.)
[2]  National Institute of Cartography, Yaounde P.O. Box 157, Cameroon; sandra_tchato77@yahoo.com
[3]  Department of Mathematical Economics and Econometrics, Omar Bongo University,
    Libreville P.O. Box 13113, Gabon; eulogekayo@gmail.com
[4]  Département de Génie Mécanique, Institut Universitaire de Technologie de L'université de Ngaoundéré,
    Ngaoundere P.O. Box 454, Cameroon; gounoupokamblaise@yahoo.fr
*   Correspondence: wouakowouakor@gmail.com (R.K.W.W.); pestalazypouth12@gmail.com (A.M.P.N.)

**Abstract:** The study of vibration isolation devices has become an emerging area of research in view of the extensive damage to buildings caused by earthquakes. The ability to effectively isolate seismic vibrations and maintain the stability of a building is thus addressed in this paper, which evaluates the effect of horizontal ground excitation on the response of a structure isolated by a coupled isolation system consisting of a non-linear damper (QZS) and a friction pendulum system (FPS). A single-degree-of-freedom system was used to model structures whose bases are subjected to seismic excitation in order to assess the effectiveness of the QZS–FPS coupling in reducing the structural response. The results obtained revealed significant improvements in structural performance when the QZS–FPS system uses a damper of optimum stiffness. A 30% reduction in displacement was recorded compared with QZS alone for two signals, one harmonic and the other stochastic. The response of the QZS–FPS system with soft stiffness to a harmonic pulse reveals amplitudes reaching around eight times those of the pulse at low frequencies and approaching zero at high frequencies. In comparison, the rigid QZS–FPS coupling has amplitudes 0.9 and 3.5 times higher than those of the harmonic signal. Thus, the resonance amplitudes observed for the QZS–FPS system are lower than those reported in other studies. This analysis highlights the performance differences between the two types of stiffness in the face of harmonic pulses, underlining the importance of the choice of stiffness in vibration management applications. The stochastic results show that on both hard and soft soils, the new QZS–FPS system causes structures to vibrate horizontally with maximum amplitudes of the order of 0.003 m and 0.007 m respectively. So, QZS–FPS coupling can be more effective than all other isolators for horizontal ground excitation. In addition, the study demonstrated that the QZS–FPS combination can offer better control of building vibration in terms of horizontal displacements.

**Keywords:** building structure; mechanical system; non-linear isolation; amplitude; damper; seismic control

## 1. Introduction

In building structures and mechanical systems, non-linear vibration isolation is considered an effective method for protecting a structure and is becoming increasingly popu-

lar [1–4]. A non-linear vibration isolation device alone or a hybrid device has the potential to significantly reduce the fundamental frequency of the protected structure to minimize the transmission of force or displacement even for low-frequency excitation cases, where traditional vibration isolations are generally not very effective [5].

Passive control devices are systems that do not require an external power source. The forces developed in these devices due to building movements are used by the devices themselves. These devices include basic isolation systems and tuned mass dampers (TMDs), quasi-zero stiffness (QZS) systems, and friction pendulum systems (FPSs) [6,7]. Previous research has demonstrated the importance of isolation bearings as they play a key role in facilitating the transmission of forces from the superstructure to the substructure [3,8–10]. The QZS is a system generally used to damp vertical forces and its effectiveness has been demonstrated on several occasions by researchers [10–15]. This quasi-zero stiffness system has been studied by numerous authors, who have presented it either as a combination of springs with positive and negative stiffness close to zero overall [16] or as a spring with negative stiffness when the latter is almost zero [9,17]. Research on QZS dampers has been conducted to reduce earthquake-induced building movements. Several studies have evaluated the responses of QZS systems under both harmonic and random excitations [17,18].

The friction pendulum system (FPS) is frequently used in civil engineering structures for isolation purposes. Due to its geometrical design, the FPS provides an effective friction isolation system by combining sliding motion with restoring force [19–23]. Some studies have evaluated the seismic reliability of an FPS-isolated base structure by treating isolator characteristics and principal earthquake characteristics as independent random variables [24].

For greater efficiency, isolation systems have often been combined with NS–TMD [25], NS–FPS [26], FPS–TMD, FPSIS [27], and QZS–inerter. In this study, we combine quasi-zero stiffness and frictional pendulum systems (QZS–FPS). In this work, the QZS used is an isolator that combines positive and negative stiffness to achieve near-zero stiffness. Combined with the FPS, the aim is to create a system with low stiffness around the equilibrium point, effectively isolating the more destructive low-frequency vibrations and dissipating energy by binding residual displacements. Building on previous work carried out on the QZS and FPS individually, this analytical and numerical study of the QZS–FPS combination provides a non-linear isolator of the horizontal ground excitation, more effective than either of them individually. The amplitude and time response curves were used respectively to assess the stability and performance of the new system in the frequency domain and to analyze the dynamic behavior of the isolated structures over time.

## 2. Equation of Motion

*2.1. Equation of Motion for the Case of Quasi-Zero Stiffness*

Figure 1 shows a QZS-isolated structure subjected to a horizontal external excitation ($\overrightarrow{\ddot{u}}_g$) and whose response $\overrightarrow{u}_b$ is determined from Equation (1).

$$\overrightarrow{P}_{base} + \overrightarrow{R}_{sol} + \overrightarrow{F}_{armo} + \overrightarrow{T}_{elon} + \overrightarrow{F}_z = mb\overrightarrow{a}_b \tag{1}$$

where:

$\overrightarrow{P}_{base}$ : the weight of the base;

$\overrightarrow{R}_{sol}$: the reaction of the ground due to the load of the building;

$\overrightarrow{F}_{armo}$: the inherent damping force of the base conferred on it by the various materials of which it is made;

$\overrightarrow{F}_z$: the force of the non-linear elastic damper;

$\vec{T}_{elon}$: the elongation force of the structure imparted by the steel;

$\vec{a_b}$: the acceleration felt at the base (foundation);

mb: the mass of the base above the isolation system;

$\gamma$: the damping ratio of the non-linear damper to that of the structure;

$k$: the ratio of the stiffness of the damper structure to that of the structure.

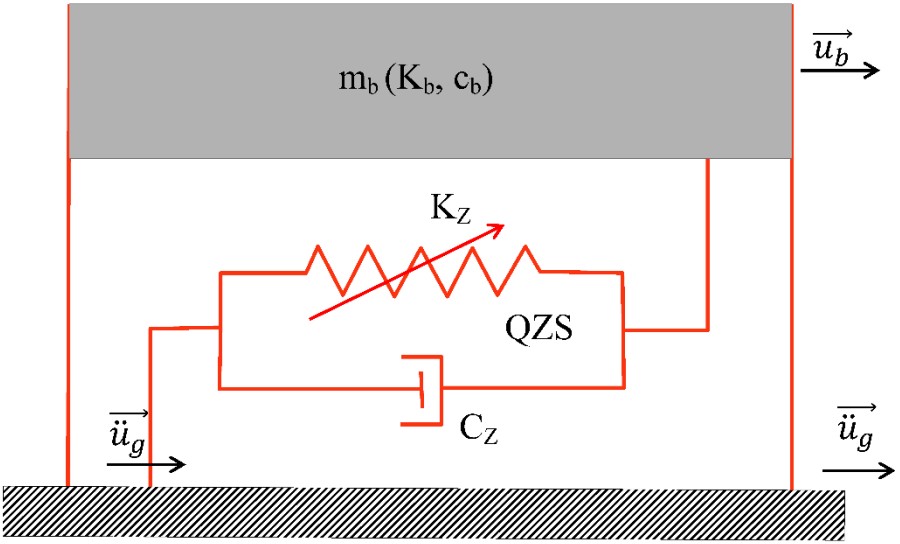

**Figure 1.** Structure isolated by QZS under horizontal excitation.

By replacing each term in Equation (1) by its value, Equation (2) is obtained.

$$c_b \dot{u}_b + k_z u_b^3 + c_z \dot{u}_b + k_b u_b = -mb.\left(\ddot{u}_b + \ddot{u}_g\right) \tag{2}$$

Expanding Equation (2) gives Equation (3)

$$\ddot{u}_b + \left(2\varepsilon_b \omega_b \dot{u}_b + 2\varepsilon_b \omega_b \gamma \dot{u}_b\right) + k\omega_b^2 u_b^3 + \omega_b^2 u_b = -\ddot{u}_g \tag{3}$$

where:

$$\varepsilon_b = \frac{c_b}{2mb\omega_b} \tag{4}$$

$$\omega_b = \sqrt{\frac{k_b}{mb}} \tag{5}$$

$$\gamma = \frac{c_z}{c_b} \tag{6}$$

$$k = \frac{k_z}{k_b} \tag{7}$$

The dimensionless form of the equation of motion is obtained from the time scale Equation (8)

$$\tau = \omega_b t \tag{8}$$

and taking the excitation signal, Equation (9)

$$\ddot{u}_g = a_0 X_0(\tau) \tag{9}$$

where $a_0$ is the seismic intensity scale, which has the same dimension as the signal acceleration, and $X_0(\tau)$ is the time non-dimensional function describing the excitation signal history.

By integrating Equations (4)–(9), Equation (10) is obtained

$$\ddot{\varphi}_b + 2(1+\gamma)\varepsilon_b \dot{\varphi}_b + \varphi_b + k\frac{a_0^2}{\omega_b^4}\varphi_b^3 = X_0(\tau) \tag{10}$$

where:

$$\varphi_b = \frac{\omega_b^2}{a_0}u_b \tag{11}$$

For the scaled function $X_0(\tau)$ harmonic, Equation (12)

$$X_0(\tau) = X_0\sin(\beta\tau + \varnothing) \tag{12}$$

and using the harmonic balance [28], the response of the structure will be expressed as Equation (13)

$$\varphi_b = X_b\sin(\beta\tau) \tag{13}$$

with β the scaled pulsation of the signal used in Equation (14)

$$-X_b\beta^2\sin(\beta\tau) + (2\varepsilon_b(1+\gamma)X_b\beta\cos(\beta\tau)) + \left(k\frac{a_0^2}{\omega_b^4}(X_b\sin(\beta\tau))^3 + X_b\sin(\beta\tau)\right) = X_g\sin(\beta\tau + \varnothing) \tag{14}$$

However, from Equations (15) and (16)

$$\sin(\beta\tau)^3 = \frac{3}{4}\sin(\beta\tau) - \frac{1}{4}\cos(3(\beta\tau)) \tag{15}$$

$$\sin(\beta\tau + \varnothing) = \sin(\beta\tau)\cos(\varnothing) - \cos(\beta\tau)\sin(\varnothing) \tag{16}$$

And neglecting the terms in sin (3(βτ)) [18,29–34], Equation (14) becomes Equation (17).

$$-X_b\beta^2\sin(\beta\tau) + (2\varepsilon_b(1+\gamma)X_b\beta\cos(\beta\tau)) + \left(\frac{3}{4}k\frac{a_0^2}{\omega_b^4}X_b^3\sin(\beta\tau) + X_b\sin(\beta\tau)\right) = X_g(\sin(\beta\tau)\cos(\varnothing) - \cos(\beta\tau)\sin(\varnothing)) \tag{17}$$

From Equation (17), we can deduce Equations (18) and (19);

$$-X_b\beta^2 + \frac{3}{4}k\frac{a_0^2}{\omega_b^4}X_b^3 + X_b = X_g\cos(\varnothing) \tag{18}$$

$$2\varepsilon_b(1+\gamma)X_b\beta = X_g\sin(\varnothing) \tag{19}$$

Squaring Equations (18) and (19) and adding them together gives Equation (20):

$$\frac{9}{16}k^2\frac{a_0^4}{\omega_b^8}X_b^6 + \frac{3}{2}k(1-\beta^2)\frac{a_0^2}{\omega_b^4}X_b^4 + ((1-\beta^2)^2 + 4\varepsilon_b^2\beta^2(1+\gamma)^2)X_b^2 - X_g^2 = 0 \tag{20}$$

After posing Equation (21),

$$\varepsilon = \varepsilon_b(1+\gamma) \tag{21}$$

Equation (20) becomes Equation (22)

$$\frac{9}{16}k^2\frac{a_0^4}{\omega_b^8}X_b^6 + \frac{3}{2}k(1-\beta^2)\frac{a_0^2}{\omega_b^4}X_b^4 + ((1-\beta^2)^2 + 4\beta^2\varepsilon^2)X_b^2 - X_g^2 = 0 \tag{22}$$

*2.2. Equation of Motion for the Case of Quasi-Zero Stiffness Coupled to the FPS*

Figure 2 shows the structure isolated by the coupling of the QZS and the FPS. The equation of motion governing the behavior of this structure under external excitation is determined from Equation (23).

$$\vec{P}_{base} + \vec{R}_{sol} + \vec{F}_{armo} + \vec{T}_{elon} + \vec{F}_z + \vec{F}_b = mb\vec{a}_b \tag{23}$$

where:

$\vec{P}_{base}$ : the weight of the base;

$\vec{R}_{sol}$: the reaction of the ground due to the load of the building

$\vec{F}_{armo}$: the inherent damping force of the base conferred on it by the various materials from which it is made;

$\vec{F}_z$: the force of the non-linear elastic damper;

$\vec{F}_b$: the force of the double FPS;

$\vec{T}_{elon}$: the elongation force of the structure conferred on it by the steel;

$\vec{a}_b$: the acceleration felt at the base (foundation);

mb: the mass of the base above the isolation system;

$\gamma$: the ratio of the damping coefficient of the non-linear damper to that of the structure;

$k$: the ratio of the stiffness of the damper to that of the structure;

$\mu(\dot{u}_b)$: the sliding friction coefficient;

$k_b$: the stiffness constant of the structure;

$k_z$: the stiffness constant of the non-linear damper;

$f_{max}$: the maximum value of the coefficient of friction.

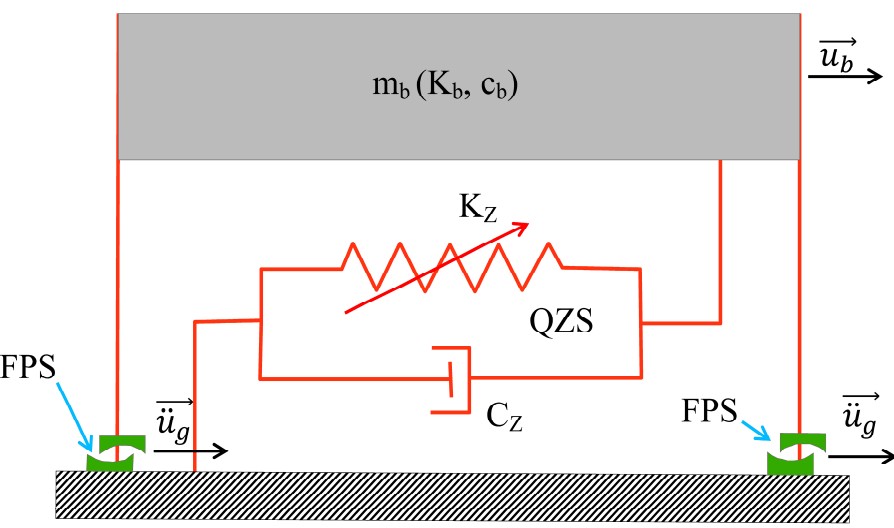

**Figure 2.** Structure isolated by QZS–FPS coupling under horizontal excitation.

By replacing each term in Equation (23) with its value, Equation (24) is obtained.

$$\ddot{u}_b + \frac{1}{mb}(c_b\dot{u}_b + k_z u_b^3 + c_z\dot{u}_b) + \frac{1}{mb}.k_b u_b + \mu(\dot{u}_b)gZ(t) = -\ddot{u}_g \tag{24}$$

From Equation (24) we deduce Equation (25):

$$\ddot{u}_b + (2\varepsilon_b\omega_b\dot{u}_b + 2\varepsilon_b\omega_b\gamma\dot{u}_b + k\omega_b^2 u_b^3) + \omega_b^2 u_b + \mu(\dot{u}_b)gZ(t) = -\ddot{u}_g \tag{25}$$

where $\varepsilon_b$, $\omega_b$, $\gamma$, and $k$ are defined by Equations (4)–(7).

Writing Equations (26) and (27)

$$Z(t) = \text{sign}\left(\dot{u}_b\right) \tag{26}$$

$$\mu\left(\dot{u}_b\right) = f_{max} \tag{27}$$

and using the principle of Equation (3), Equation (28) is obtained from Equation (25) as follows:

$$\ddot{\varphi}_b + 2(1+\gamma)\varepsilon_b\dot{\varphi}_b + \varphi_b + k\frac{a_0^2}{\omega_b^4}\varphi_b^3 + gf_{max}\text{sign}(\dot{\varphi}_b) = X_0(\tau) \tag{28}$$

Assuming harmonic seismic excitation, Equation (29)

$$X_0(\tau) = X_g\sin(\beta\tau + \varnothing) \tag{29}$$

the response of the structure, Equation (31) will be expressed as [28]:

$$\varphi_b = X_b\sin(\beta\tau) \tag{30}$$

$$\begin{aligned} -X_b\beta^2\sin(\beta\tau) \quad &+(2\varepsilon_b\omega_b X_b\beta\cos(\beta\tau) + 2\gamma\omega_b X_b\beta\cos(\beta\tau)) \\ &+(k\omega_b^2(X_b\sin(\beta\tau))^3 + \omega_b^2 X_b\sin(\beta\tau)) + f_{max}sign(X_b\beta\cos(\beta\tau)) \\ &= X_g\sin(\beta\tau + \varnothing) \end{aligned} \tag{31}$$

According to Equation (32)

$$sgn(X_b\beta\cos(\beta\tau)) = \frac{4}{\pi}\beta\cos(\beta\tau) \tag{32}$$

And neglecting the terms in $\sin(3(\beta\tau))$, Equation (31) becomes Equation (33)

$$\begin{aligned} -X_b\beta^2\sin(\beta\tau) + (2\varepsilon_b\omega_b X_b\beta\cos(\beta\tau) + 2\varepsilon_b\omega_b\gamma X_b\beta\cos(\beta\tau)) + \left(\frac{3}{4}k\omega_b^2 X_b^3\sin(\beta\tau) + \omega_b^2 X_b\sin(\beta\tau)\right) \\ +\frac{4}{\pi}f_{max}\beta\cos(\beta\tau) = X_g(\sin(\beta\tau)\cos(\varnothing) + \cos(\beta\tau)\sin(\varnothing)) \end{aligned} \tag{33}$$

From Equation (33) we obtain Equations (34) and (35):

$$(1-\beta^2)X_b + \frac{3}{4}k\frac{a_0^2}{\omega_b^4}X_b^3 = X_g\cos(\varnothing) \tag{34}$$

$$2\varepsilon_b(1+\gamma)X_b\beta + \frac{4}{\pi}f_{max}\beta = X_g\sin(\varnothing) \tag{35}$$

Squaring Equations (34) and (35) and adding them together gives Equation (36)

$$\frac{9}{16}k^2\frac{a_0^4}{\omega_b^8}X_b^6 + \frac{3}{2}k(1-\beta^2)\frac{a_0^2}{\omega_b^4}X_b^4 + ((1-\beta^2)^2 + 4\varepsilon^2\beta^2)X_b^2 + \frac{16}{\pi}\varepsilon X_b\beta^2 f_{max} + \frac{16}{\pi^2}f_{max}^2\beta^2 = X_g^2 \tag{36}$$

where $\varepsilon$ is given in Equation (21).

### 2.3. Case of Stochastic Excitation

In this case, the seismic loading $(\ddot{u}_g)$ is represented as random sequences of white Gaussian noise, adjusted by filtering and time modulation of varying intensity, in the context of spectral density analysis. The filter characteristics determine the frequency distribution of these random stresses and are adjusted to correspond to rigid, intermediate,

or loose soil conditions, as appropriate. For this purpose, the Kanai–Tajimi filter modified by Clough and Penzien (Equations (37) and (38)) is used to model the different soil types.

$$S_{\ddot{u}_g}(\beta) = \frac{\beta^4 + 4\varepsilon_g^2\omega_g^2\beta^2}{(\omega_g^2 - \beta^2)^2 + 4\varepsilon_g^2\omega_g^2\beta^2} \frac{\beta^4}{(\omega_f^2 - \beta^2)^2 + 4\varepsilon_f^2\omega_f^2\omega^2} S_w \tag{37}$$

$$S_w = \frac{0.141\varepsilon_g}{\omega_g\sqrt{1 + 4\varepsilon_g^2}}\ddot{u}_{g0}^2 \tag{38}$$

$S_w$ represents the spectral level of white noise linked to the maximum acceleration of the ground and $\ddot{u}_{g0}$ is the peak ground acceleration (PGA). $\varepsilon_g$, $\omega_g$, $\varepsilon_f$, and $\omega_f$ are the filter parameters [35,36].

According to [37–39], we assume that the maximum value of the recorded ground acceleration oscillates between 0.4 g and 0.6 g.

## 3. Results and Discussion

### 3.1. Amplitude–Frequency Response in the Case of a Harmonic Signal

Figure 3 shows the amplitude response of the non-isolated system isolated by the QZS according to its various characteristics as a function of the scaled frequency for a total system damping of 5%. In the case of a system isolated by a QZS with a softening stiffness (k = −0.0249), the amplitude curves slope as the frequency increases; in other words, the resonance frequency is inversely proportional to the amplitude of the oscillations. In contrast, for the system isolated by a rigid QZS (k = 0.01000 and k = 0.6000), the amplitude curves slope to the left, from which we deduce that the responses are weak at high resonance frequencies.

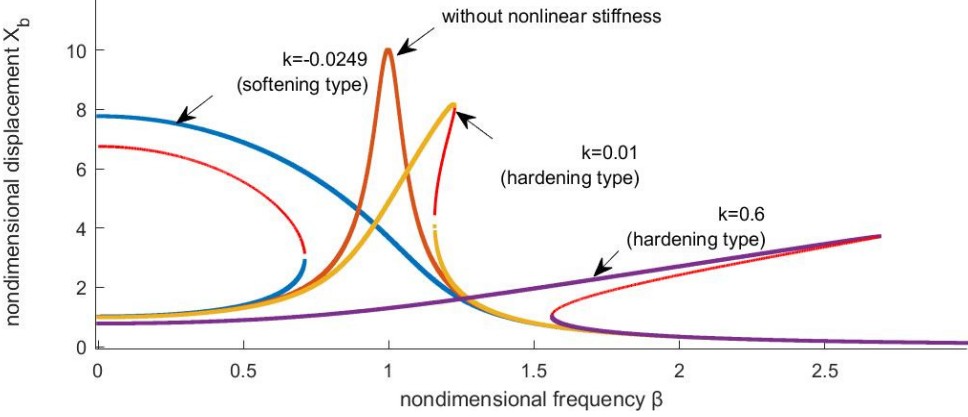

**Figure 3.** Amplitude–frequency response, case of QZS.

The amplitude–frequency responses of the system isolated by an FPS and by the QZS–FPS coupling are shown in Figure 4 for ε = 0.025. The slope of the amplitude–frequency response curves in Figure 4 shows the persistence of the QZS non-linearity despite the QZS–FPS coupling. In addition, the maximum amplitudes of the responses of the structure for the case of soft–stiff QZS coupling with the FPS are greater than those of the structure isolated only with an FPS. This is quite the opposite of those where the structure is isolated by a rigid QZS coupling and an FPS, which are the lowest as the stiffness increases.

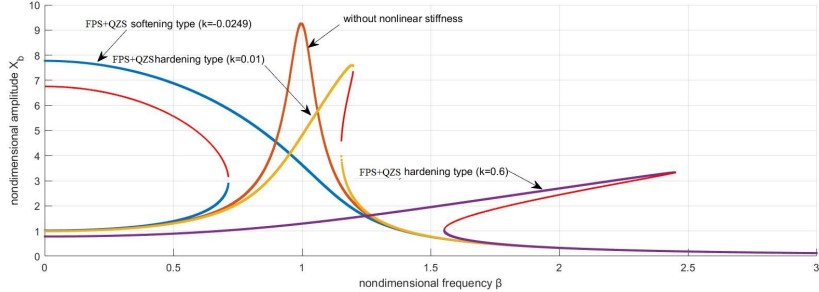

**Figure 4.** Amplitude–frequency response, case of QZS–friction pendulum coupling.

### 3.2. Amplitude Response for the Case of a Stochastic Signal

The various seismic signals were derived from Equation (17).

#### 3.2.1. Case of Soft Ground

Figure 5 shows the time evolution of the seismic signal Xg, modeled according to the Kanai–Tajimi spectrum modified by Clough and Penzien. The signal shows amplitude variations between $-4 \, \text{m/s}^2$ and $3 \, \text{m/s}^2$ over a period of 40 s. The oscillations observed are characterized by acceleration and deceleration phases, which are essential for analyzing the dynamic response of structures.

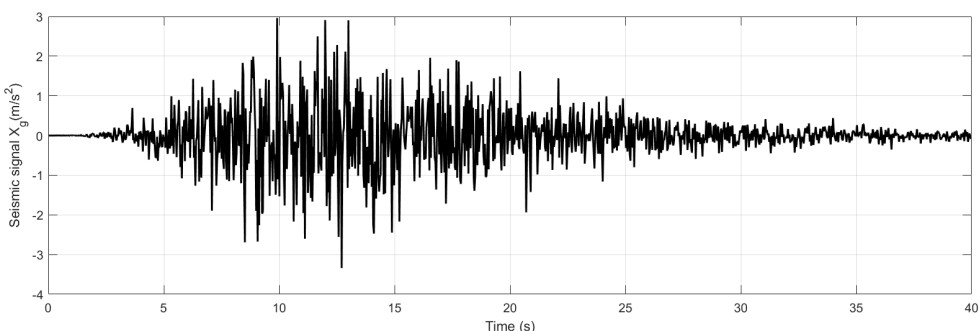

**Figure 5.** Seismic signal for soft ground.

The response of the structure to this signal is shown in Figure 6 for the different cases of isolation (QZS and QZS–FPS coupling). For an uninsulated structure, the amplitudes vary between $-0.028$ m and $0.026$ m, while in the case of low stiffness isolation, the amplitudes of the oscillations vary between $-0.021$ m and $0.019$ m. The negative and positive displacements show that the building oscillates around its equilibrium position. Furthermore, the amplitudes of the oscillations in the case of rigid stiffness isolation vary between $-0.021$ m and $0.019$ m. In other words, the amplitudes of the oscillations in the case of isolation with a soft stiffness are smaller than those of isolation with a rigid stiffness.

The response of the structure to the seismic signal in a soft ground environment is shown in Figure 7. The amplitude of the oscillations for the case of a QZS coupling with soft-stiffness FPS varies between $-0.0052$ m and $0.0071$ m while that isolated by the friction pendulum alone varies between $-0.0094$ m and $0.0065$ m. Conversely, Figure 7b shows that for a high-stiffness damper, the amplitude curve varies between $-0.0056$ m and $0.0069$ m.

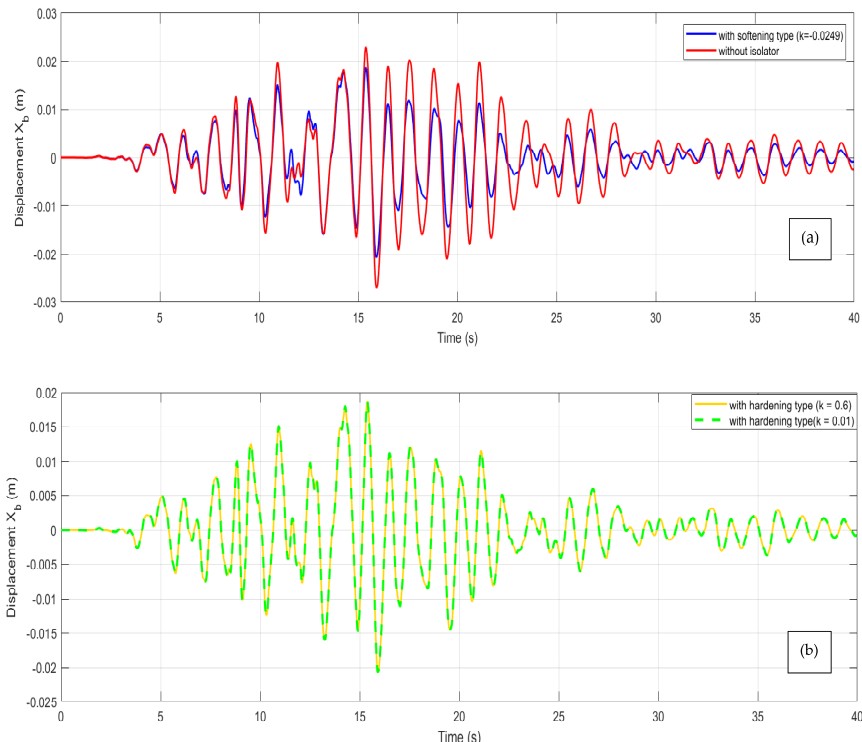

**Figure 6.** Response of the structure on soft soil: (**a**) case of a non-isolated structure and isolated by a soft-stiffness QZS; (**b**) case of a structure isolated by a rigid-stiffness QZS (k = 0.01 and k = 0.6).

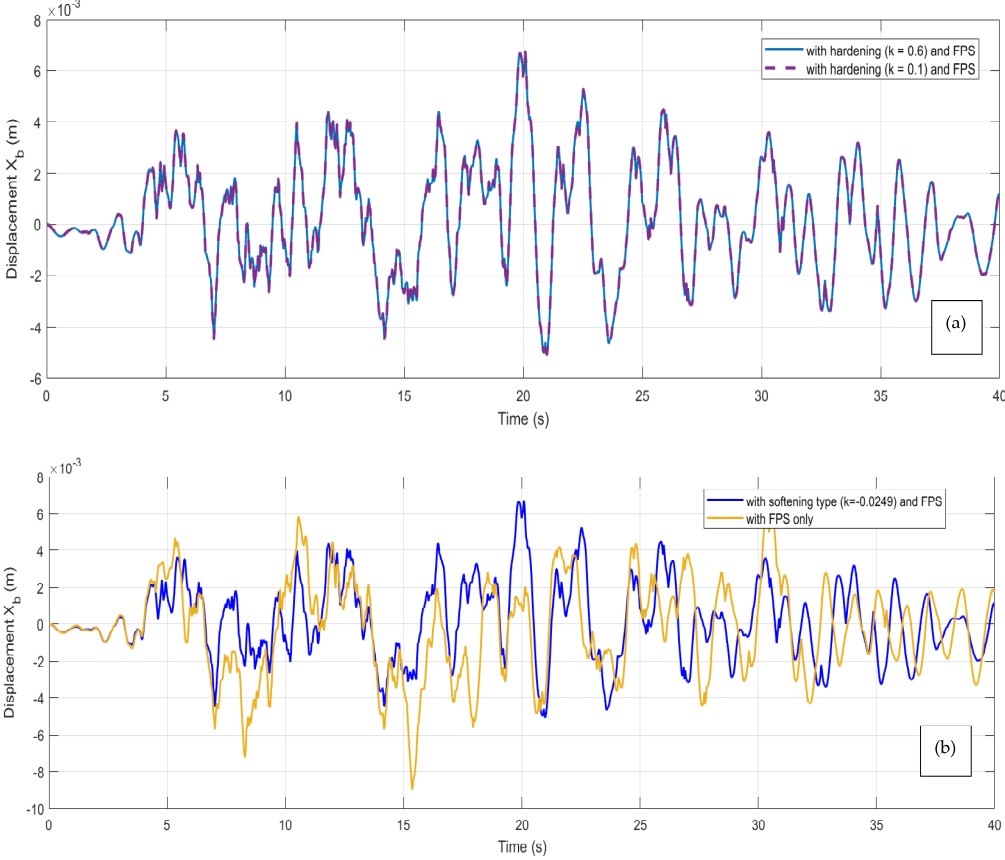

**Figure 7.** Response of the structure to the seismic signal on soft soil: (**a**) case of a structure isolated by the soft-stiffness QZS-FPS coupling; (**b**) case of a structure isolated by the rigid-stiffness QZS–FPS coupling (k = 0.01 and k = 0.6).

### 3.2.2. Case of Hard Ground

Figure 8 shows the temporal evolution of the Xg seismic signal in a rigid soil environment.

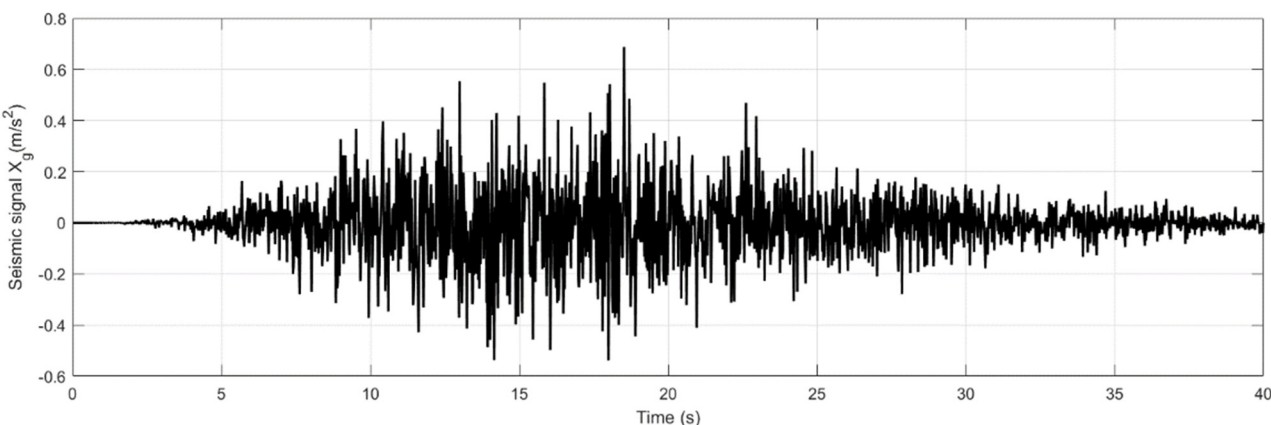

**Figure 8.** Seismic signal for hard ground.

Figure 9 shows the time trace of the response of the structure. For the same seismic signal, the amplitudes are constant to a few decimal places in the case of a rigid QZS, whereas the amplitudes of the oscillations decrease in the case of a soft QZS.

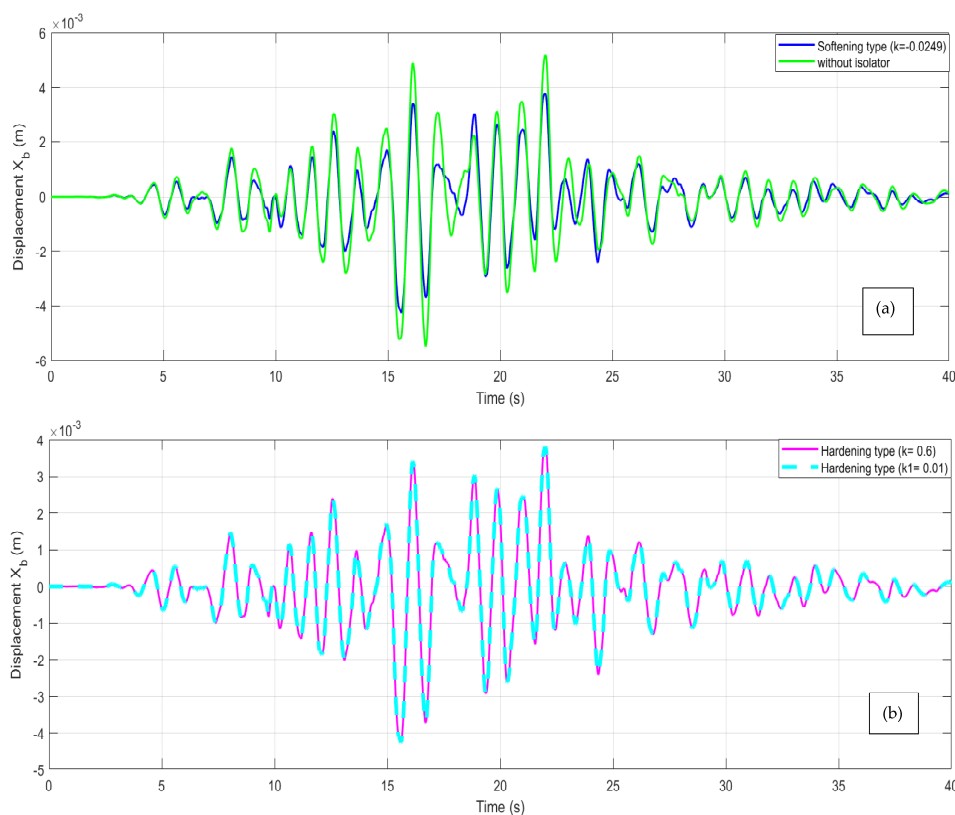

**Figure 9.** Response of the structure on hard ground: (**a**) case of a non-isolated structure isolated by the soft-stiffness QZS; (**b**) case of a structure isolated by the rigid-stiffness QZS (k = 0.01 and k = 0.6).

Figure 10 shows the response of the structure under different types of isolation. The blue curve in Figure 10a shows the evolution of the dependent variable (for example, performance or displacement) in a system where a softening with a coefficient k = −0.0249 is applied, in conjunction with FPS management, whereas the green curve shows the

evolution of the same dependent variable in a system where only FPS management is applied. As for Figure 10b, the interrupted curve represents the response of the structure isolated by the QZS–FPS coupling with rigid stiffness k = 0.6 and the solid line represents the QZS–FPS coupling with rigid stiffness k = 0.01. Figure 6 shows that the structure isolated by the soft–stiff QZS–FPS coupling has a lower amplitude than that isolated by the QZS and the stiff QZS–FPS coupling.

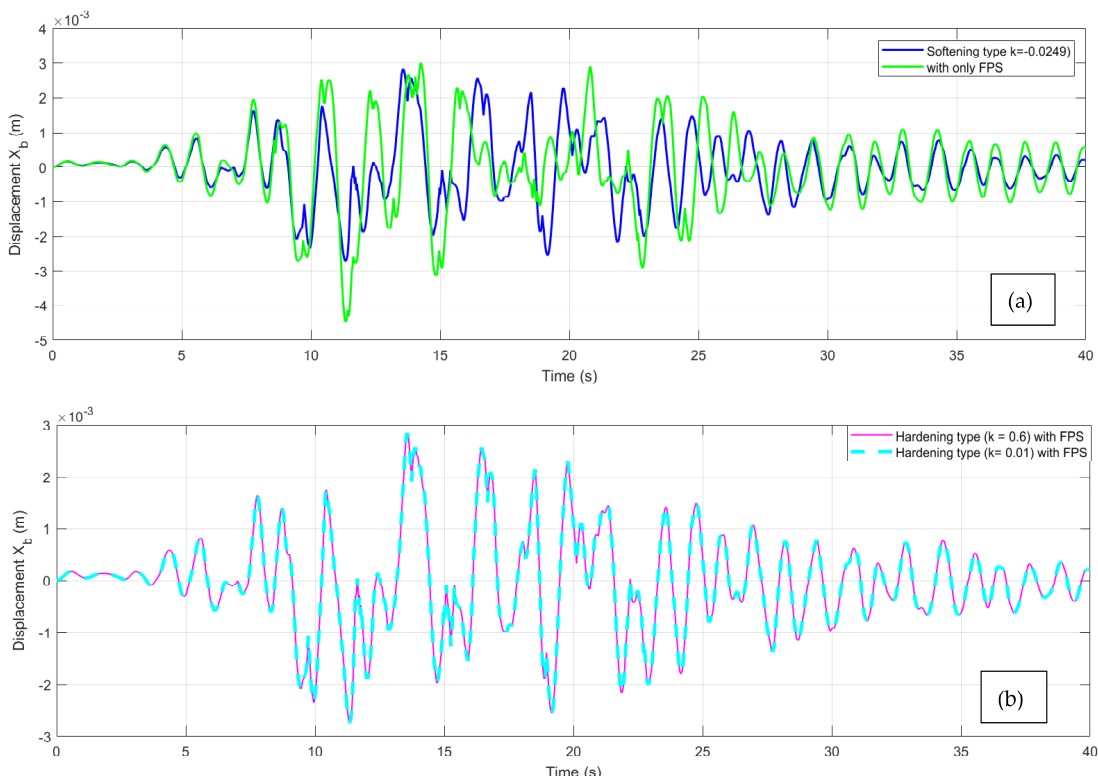

**Figure 10.** Response of the isolated structure: (**a**) by the FPS and by the coupling (QZS with soft stiffness and FPS); (**b**) by the coupling QZS–FPS with rigid stiffnesses k = 0.6 and k = 0.01.

### 3.3. Discussion

#### 3.3.1. Case of a Harmonic Signal

Figures 3 and 4 show the amplitude–frequency response of a structure that was initially uninsulated, then isolated by the QZS and by the QZS–FPS coupling at different stiffnesses. The curves observed in Figure 4 show the importance of the non-linearity of the QZS compared with that of the friction pendulum [16,35]. The response of the soft–stiff QZS–FPS system to a harmonic pulse showed amplitudes of the order of eight times that of the excitation pulse for low frequencies and amplitudes close to 0 for high frequencies. In the case of the QZS–FPS system with rigid stiffness, the oscillations have amplitudes of the order of 0.9 and 3.5 times those of the harmonic signal, but lower than those of [40,41], showing that the FPS plays a major role not only in the performance but also in the stabilization of the system and that the QZS–FPS system with soft stiffness does not react very effectively to low-frequency harmonic signals, unlike that with rigid stiffness.

#### 3.3.2. Case of a Stochastic Signal

The effectiveness of the proposed solution is illustrated by a decrease in oscillation amplitudes of around 30% on soft and hard ground in Figures 7 and 10. In addition, Figures 7 and 10 show that the QZS–FPS coupling with soft stiffness is the most effective under different types of soil, probably due to the performance of the QZS with soft stiffness,

as shown in Figure 6a and [4,42]. Moreover, on both hard and soft ground, the new QZS–FPS system causes structures to vibrate horizontally with maximum amplitudes of the order of 0.003 m and 0.007 m, respectively. Finally, when compared with pre-existing hybrid dampers (Table 1), the QZS–FPS system outperformed NS–TMD, FPSIS, NS–FPS, and FPS–TMD.

**Table 1.** Comparison of the performance of hybrid systems proposed by various authors.

| Systems | Improvements | Reference |
|---|---|---|
| NS–TMD | Improves the stability of a bridge previously equipped with FPS. Reduces vibration with a maximum output amplitude of 0.081 m. | [25] |
| FPSIS | Reduces vibrations with a maximum amplitude of 0.8 m. | [27] |
| NS–FPS | Presents an innovative solution to improve the seismic resilience of above-ground buildings and underground infrastructures (ASUS) by coupling negative stiffness and friction pendulum. | [26] |
| FPS–TMD | Reduces vibration with a maximum amplitude of 0.1 m. | [27] |

## 4. Conclusions

Non-linear isolation devices are an emerging area of research and have, due to their promising potential, attracted considerable attention in the scientific community. The aim of this study was to evaluate the effect of horizontal ground excitation on the response of a structure isolated by a coupled system consisting of a non-linear damper (QZS) and a friction pendulum (FPS), using harmonic signal and stochastic signal. Analysis of the response of the QZS–FPS system with soft and rigid stiffness reveals its remarkable ability to absorb harmonic pulses. However, the QZS–FPS system with soft stiffness does not respond very effectively to low-frequency harmonic signals, whereas the system with rigid stiffness does. This distinction underlines the importance of choosing the type of stiffness according to specific application requirements. The stochastic results show that on both hard and soft soils, the proposed QZS–FPS system causes structures to vibrate horizontally with maximum amplitudes of the order of 0.003 m and 0.007 m, respectively. So, the QZS–FPS coupling can be more effective than all other isolators for horizontal ground excitation. To improve on the existing work, another study with several degrees of freedom could be envisaged.

**Author Contributions:** Conceptualization, R.K.W.W. and P.N.N.; methodology, R.K.W.W., S.C.T., E.F.K.P., B.P.G.P., A.M.P.N., E.M.D. and P.N.N.; software, R.K.W.W., S.C.T., E.F.K.P., B.P.G.P., A.M.P.N., E.M.D. and P.N.N.; validation, R.K.W.W., S.C.T., E.F.K.P., B.P.G.P., A.M.P.N., E.M.D. and P.N.N.; formal analysis, R.K.W.W., A.M.P.N. and P.N.N.; investigation, R.K.W.W., A.M.P.N. and P.N.N.; data curation, R.K.W.W., S.C.T., E.F.K.P., B.P.G.P., A.M.P.N., E.M.D. and P.N.N.; validation, R.K.W.W., S.C.T., E.F.K.P., B.P.G.P., A.M.P.N., E.M.D. and P.N.N.; writing—original draft preparation, R.K.W.W., S.C.T., E.F.K.P., B.P.G.P., A.M.P.N., E.M.D. and P.N.N.; validation, R.K.W.W., S.C.T., E.F.K.P., B.P.G.P., A.M.P.N., E.M.D. and P.N.N.; writing—review and editing, R.K.W.W., S.C.T., E.F.K.P., B.P.G.P., A.M.P.N., E.M.D. and P.N.N.; validation, R.K.W.W., S.C.T., E.F.K.P., B.P.G.P., A.M.P.N., E.M.D. and P.N.N.; visualization, R.K.W.W. and E.M.D.; supervision, P.N.N.; project administration, E.M.D. and P.N.N.; All authors have read and agreed to the published version of the manuscript.

**Funding:** This research received no external funding.

**Data Availability Statement:** The data presented in this study are available on request from the corresponding author.

**Acknowledgments:** The authors would like to thank the editors, the three anonymous reviewers for their professional comments to improve the manuscript and also Ngo Nouck Hélène, Nguentchue

Rose, Meli'i Jorelle Larissa, Department of Physics, University of Yaounde I, Mbouombouo Ngapouth Ibrahim, Department of Physics, University of Yaounde I, Marthe Ariane Mbond Gweth, National Advanced School of Public Work, Yaounde Cameroon, Fotsa Tematio Patrice, Civil Engineer at UCB S.A. Cameroon and Nongni Tsopmo F. for the facilitation.

**Conflicts of Interest:** The authors declare no conflicts of interest.

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
