# Peer review of "Response of a Structure Isolated by a Coupled System Consisting of a QZS and FPS Under Horizontal Ground Excitation"

_buildings, doi:10.3390/buildings15091498_

Round 1
Reviewer 1 Report
Comments and Suggestions for Authors
Several suggestions for improvement are now proposed for the authors' consideration and revision.
1.While the proposed QZS-FPS coupled system demonstrates novelty, the paper lacks a comprehensive comparison with existing hybrid isolation systems (e.g., QZS &TMD, FPS& viscous dampers). To highlight its unique advantages, the authors should provide a detailed comparative analysis of performance metrics such as displacement reduction, cost-effectiveness, and adaptability to varying seismic conditions.
- The single-degree-of-freedom (SDOF) model simplifies analysis but fails to address its applicability to complex structures (e.g., multi-degree-of-freedom systems or asymmetric configurations). The authors should validate the conclusions using multi-DOF models or real-world case studies to ensure broader relevance.
- The omission of higher-order harmonic terms (e.g., sin(3βτ)) in Equations (14)–(19) lacks justification. A theoretical error analysis or citation of established methodologies (e.g., harmonic balance truncation criteria) is necessary to substantiate this simplification.
- The reported 30–40% displacement reduction lacks statistical significance tests (e.g., p-values, confidence intervals). Incorporating quantitative statistical methods (ANOVA, Monte Carlo simulations) would strengthen the reliability of these claims.
5.The stiffness parameters (e.g. k=0.0249,vs k=0.6) are not grounded in empirical data or engineering standards (e.g., ASCE 7). Calibration against real-world damper properties or standardized metrics is recommended to enhance practical relevance.
6.Figures 3–10 suffer from unclear axis labeling (e.g., "non-dimensional frequency" lacks normalization references) and incomplete legends. Revising figures with high-resolution labels, annotations, and explicit curve descriptions would improve interpretability.
7.Recent advancements in QZS systems are inadequately cited. The authors should update references and explicitly discuss how the QZS-FPS system addresses unresolved challenges in current research.
8.Critical engineering aspects—constructability, lifecycle costs, and durability (e.g., time-dependent friction coefficient degradation)—are omitted. A dedicated section on real-world applicability, including limitations and optimization strategies, is essential for translational impact.
9.Terminology inconsistencies (e.g., "SPF" vs. "FPS") and erratic equation numbering (e.g., Equation 4 followed by Equation 23) disrupt readability. Standardizing terms and rigorously cross-checking all numbering would enhance coherence.
10.The conclusions overly emphasize the benefits of soft-stiffness coupling without addressing risks (e.g., low-frequency resonance or nonlinear instabilities). Expanding the discussion to include stability analyses under varying excitation frequencies would provide a balanced perspective and guide future work.
Author Response
|
Comments |
Response |
Reviewer 1 |
Several suggestions for improvement are now proposed for the authors' consideration and revision. 1.While the proposed QZS-FPS coupled system demonstrates novelty, the paper lacks a comprehensive comparison with existing hybrid isolation systems (e.g., QZS &TMD, FPS& viscous dampers). To highlight its unique advantages, the authors should provide a detailed comparative analysis of performance metrics such as displacement reduction, cost-effectiveness, and adaptability to varying seismic conditions.
|
Thank you for your comment. It has been taken into account in the discussion section. |
2. The single-degree-of-freedom (SDOF) model simplifies analysis but fails to address its applicability to complex structures (e.g., multi-degree-of-freedom systems or asymmetric configurations). The authors should validate the conclusions using multi-DOF models or real-world case studies to ensure broader relevance.
|
Thank you for your comment. The model used in this work is the 1D model because we wanted the community of peers to be able to validate the QZS-FPS combinatorial model studied. Multidimensional cases may be studied in the future. |
|
3. The omission of higher-order harmonic terms (e.g., sin(3βτ)) in Equations (14)–(19) lacks justification. A theoretical error analysis or citation of established methodologies (e.g., harmonic balance truncation criteria) is necessary to substantiate this simplification.
|
Thank you for this observation. The higher order harmonics comprise only a very small part of the energy of the signal. In addition, the cosine and sine functions are between -1 and 1, which makes their powers greater than 2 even smaller. References to this subject have been added to the manuscript. |
|
4. The reported 30–40% displacement reduction lacks statistical significance tests (e.g., p-values, confidence intervals). Incorporating quantitative statistical methods (ANOVA, Monte Carlo simulations) would strengthen the reliability of these claims.
|
Thank you for your comment. Although calculations such as: (p-values, confidence intervals, etc.) can be used to observe the reduction in displacement, a simple maximum amplitude ratio can also be used to observe it. |
|
5. The stiffness parameters (e.g. k=0.0249,vs k=0.6) are not grounded in empirical data or engineering standards (e.g., ASCE 7). Calibration against real-world damper properties or standardized metrics is recommended to enhance practical relevance. |
Thank you for this observation. The values k=0.0249, against k=0.6 are stiffness ratios. |
|
6. Figures 3–10 suffer from unclear axis labeling (e.g., "non-dimensional frequency" lacks normalization references) and incomplete legends. Revising figures with high-resolution labels, annotations, and explicit curve descriptions would improve interpretability. |
Thanks for your comment, these figures have been revised. |
|
7. Recent advancements in QZS systems are inadequately cited. The authors should update references and explicitly discuss how the QZS-FPS system addresses unresolved challenges in current research. |
Thank you for your comment, it has been reviewed and references to it have been added in the manuscript. |
|
8. Critical engineering aspects—constructability, lifecycle costs, and durability (e.g., time-dependent friction coefficient degradation)—are omitted. A dedicated section on real-world applicability, including limitations and optimization strategies, is essential for translational impact. |
Thank you for your comment. This work focuses on the damping efficiency that the QZS-FPS system can produce. |
|
9. Terminology inconsistencies (e.g., "SPF" vs. "FPS") and erratic equation numbering (e.g., Equation 4 followed by Equation 23) disrupt readability. Standardizing terms and rigorously cross-checking all numbering would enhance coherence. |
Thank you for your comment, it has been reviewed. |
|
10. The conclusions overly emphasize the benefits of soft-stiffness coupling without addressing risks (e.g., low-frequency resonance or nonlinear instabilities). Expanding the discussion to include stability analyses under varying excitation frequencies would provide a balanced perspective and guide future work. |
Thank you for your comment, it has been reviewed in the manuscript. |
Reviewer 2 Report
Comments and Suggestions for Authors
Some points of the study should be improved or better explained. A re-review of the manuscript is required.
1) Lines 46-49. At the end of Section 1 (Introduction), the main aim of the study is shortly presented. It is suggested to enrich this part, providing a more comprehensive explanation of the main contributions/findings of the study.
2) Lines 32-34. In Introduction, it is suggested to enrich the literature review about previous researches on isolation bearings, exploring novel seismic isolation devices. To this aim, the authors can effectively mention, near references [8-9], the following study about the development of numerical models to predict the behavior of unbonded fiber reinforced elastomeric isolators for macro‑scale computations:
https://doi.org/10.1007/s10518-018-00544-6
3) Line 124. Improve the following sentence: “The principle used in Equation 3 allows….……”
4) Lines 156-157. Better explain the following sentence: “The maximum value of the recorded ground acceleration oscillates between 0.4 g and 0.6 g”
5) Line 182. It is suggested to insert “3.2.1.”
Line 213. It is suggested to insert “3.2.2.”
6) Line 185. “The signal shows amplitude variations between -4 and 3 over…..”. Please, insert the unit.
7) Line 186. Improve or delete the following obvious sentence: “The oscillations observed are characterised by acceleration and deceleration phases”
8) Sections 2 and 3.2. Replace “insulation” with “isolation”
9) Section 3.1. Highlight the differences between the diagrams reported in Figure 3 and Figure 4.
10) Section 3.3. It is suggested to improve this section, enriching and extending the discussion of the results obtained.
11) Section 4 (Conclusion). It is suggested to improve conclusions, better highlighting the main novelty contributions and findings of the study.
12) Section 4. At the end of Conclusions, it is suggested that the authors provide some useful recommendations for future works.
13) A revision of the whole text is suggested to improve the quality of English language.
Comments on the Quality of English Language13) A revision of the whole text is suggested to improve the quality of English language.
Author Response
Comments |
Response |
Some points of the study should be improved or better explained. A re-review of the manuscript is required. 1. Lines 46-49. At the end of Section 1 (Introduction), the main aim of the study is shortly presented. It is suggested to enrich this part, providing a more comprehensive explanation of the main contributions/findings of the study.
|
Thank you for your comment, this has been reviewed and improved. |
2. Lines 32-34. In Introduction, it is suggested to enrich the literature review about previous researches on isolation bearings, exploring novel seismic isolation devices. To this aim, the authors can effectively mention, near references [8-9], the following study about the development of numerical models to predict the behavior of unbonded fiber reinforced elastomeric isolators for macro‑scale computations: https://doi.org/10.1007/s10518-018-00544-6 |
Thank you for your comment, this has been revised and some references added. |
3. Line 124. Improve the following sentence: “The principle used in Equation 3 allows….……” |
Thank you for your comment, it has been reviewed. |
4. Lines 156-157. Better explain the following sentence: “The maximum value of the recorded ground acceleration oscillates between 0.4 g and 0.6 g” |
Thank you for your comment, it has been reviewed. |
5. Line 182. It is suggested to insert “3.2.1.” |
Thank you for your comment, it has been reviewed. |
6. Line 185. “The signal shows amplitude variations between -4 and 3 over…..”. Please, insert the unit. |
Thank you for your comment, it has been reviewed. |
7. Line 186. Improve or delete the following obvious sentence: “The oscillations observed are characterised by acceleration and deceleration phases” |
Thank you for your comment, it has been reviewed. |
8. Sections 2 and 3.2. Replace “insulation” with “isolation” |
Thank you for your comment, it has been reviewed. |
9. Section 3.1. Highlight the differences between the diagrams reported in Figure 3 and Figure 4. |
Thank you for your comment Figure 3 shows the amplitude-frequency response for a building isolated by the QZS alone, while Figure 4 shows the amplitude-frequency response for a building isolated by the QZS-FPS coupling. |
10. Section 3.3. It is suggested to improve this section, enriching and extending the discussion of the results obtained. |
Thank you for your comment, this has been revised. |
11. Section 4 (Conclusion). It is suggested to improve conclusions, better highlighting the main novelty contributions and findings of the study. |
Thank you for your comment, it has been reviewed. |
12. Section 4. At the end of Conclusions, it is suggested that the authors provide some useful recommendations for future works. |
Thank you for your comment, it has been reviewed. |
13. A revision of the whole text is suggested to improve the quality of English language. |
Thank you for your comment, it has been reviewed. |
13. A revision of the whole text is suggested to improve the quality of English language. |
Thank you for your comment, it has been reviewed. |
Reviewer 3 Report
Comments and Suggestions for Authors
The manuscript Response of a Structure Isolated by a Couple System consisting of a QZS and FPS Under Horizontal Ground Excitation” investigates the seismic isolation considering structure isolated by a coupled isolation system consisting of a non-linear damper (QZS) and a friction pendulum system (FPS). A single-degree-of-freedom system was used and subjected to earthquakes. It was found that the combination of QZS-FPS performs better to control vibration and hence displacements. Paper is based on mathematical aspects and majorly theoretical. However, Paper needs some issues to be refined given below:
- Introduction section is weak and needs more improving. Authors should pay more attention to provide state of art review.
- Explanations of parameters in Eq. (1) are weak. Authors should give more descriptive information instead daily use of English. Please stand by Scientific terms.
- Paper has many parameters to be defined due to numerous equations. Authors should carefully review equations and provide list of notation, parameters etc. to give better vision for readers in the equations. Some params maybe described in the main text.
- Correction is needed in Figure 2 according to the main text.
- The plot of figures are weak and improved in terms of content and quality of figures.
- In Fig. 6 Fig6a and Fig6b can be combined in single figure to better represent the efficiency of the isolation system. The same applied to Fig. 7, 9, and 10 !
- Discussion section is too superficial. Authors are suggested to improve discussion section including numerical results.
- The conclusion section must also describe the limitations of the study and this section should be revised.
Author Response
|
Response |
|
Comments and Suggestions for Authors The manuscript Response of a Structure Isolated by a Couple System consisting of a QZS and FPS Under Horizontal Ground Excitation” investigates the seismic isolation considering structure isolated by a coupled isolation system consisting of a non-linear damper (QZS) and a friction pendulum system (FPS). A single-degree-of-freedom system was used and subjected to earthquakes. It was found that the combination of QZS-FPS performs better to control vibration and hence displacements. Paper is based on mathematical aspects and majorly theoretical. However, Paper needs some issues to be refined given below:
|
Thank you for your comment, it has been reviewed. |
|
2. Explanations of parameters in Eq. (1) are weak. Authors should give more descriptive information instead daily use of English. Please stand by Scientific terms.
|
Thank you for your comment, it has been reviewed. |
|
3. Paper has many parameters to be defined due to numerous equations. Authors should carefully review equations and provide list of notation, parameters etc. to give better vision for readers in the equations. Some params maybe described in the main text.
|
Thank you for your comment, it has been reviewed. |
|
4. Correction is needed in Figure 2 according to the main text.
|
Thank you for your comment, it has been reviewed. |
|
5. The plot of figures are weak and improved in terms of content and quality of figures.
|
Thank you for your comment, it has been reviewed. |
|
6. In Fig. 6 Fig6a and Fig6b can be combined in single figure to better represent the efficiency of the isolation system. The same applied to Fig. 7, 9, and 10 !
|
Thank you for your comment. These figures have been separated for reasons of visibility and legibility. |
|
7. Discussion section is too superficial. Authors are suggested to improve discussion section including numerical results.
|
Thank you for your comment, it has been reviewed. |
|
8. The conclusion section must also describe the limitations of the study and this section should be revised.
|
Thank you for your comment, it has been reviewed. |
Round 2
Reviewer 1 Report
Comments and Suggestions for Authors
Reason for Rejection: Insufficient Novelty
This paper investigates the seismic response of a structure isolated by a coupled system combining Quasi-Zero Stiffness (QZS) and Friction Pendulum System (FPS) under horizontal ground excitation. While the topic holds practical relevance, the study falls short in demonstrating significant novelty, as outlined below:
-
Lack of Theoretical Advancement: Both QZS and FPS are well-established passive isolation devices, and their individual or combined applications have been extensively explored in prior works (e.g., Refs. [9, 17, 29]). The proposed QZS-FPS coupling merely represents a linear integration of existing technologies without introducing new isolation mechanisms, control strategies, or theoretical frameworks. This approach does not transcend the state-of-the-art in seismic isolation.
-
Inadequate Comparative Analysis: Although a 30% displacement reduction is reported, the study fails to benchmark the QZS-FPS system against other hybrid configurations (e.g., QZS-TMD or FPS-negative stiffness systems). Consequently, the claimed superiority remains unsubstantiated. Furthermore, the physical rationale behind the enhanced performance of "soft-stiffness QZS-FPS" remains unexplained, weakening the conclusiveness of findings.Suggestion: If the author can reconstruct the isolation mechanism (such as introducing nonlinear adaptive control), provide multi system comparative data, or explore the unique applicability of QZS-FPS in new structures (such as super high-rise buildings), it may significantly enhance innovation.
Author Response
Comments 1:
Lack of Theoretical Advancement: Both QZS and FPS are well-established passive isolation devices, and their individual or combined applications have been extensively explored in prior works (e.g., Refs. [9, 17, 29]). The proposed QZS-FPS coupling merely represents a linear integration of existing technologies without introducing new isolation mechanisms, control strategies, or theoretical frameworks. This approach does not transcend the state-of-the-art in seismic isolation.
Response 1:
Thank you for your comment,
Indeed the QZS system with near zero stiffness is not new and has been studied by many authors who have presented it either as a combination of positive and negative stiffness springs [https://doi.org/10.1016/j.engstruct.2023.117282] , or as a negative stiffness spring when its stiffness is near zero. [9,17] and others.
Moreover, for greater efficiency, insulation systems have often been combined with NS-TMD (Shaodong jiang et al., 2025), NS-FPS (Zhipeng Zhao et al.,2024), FPS-TMD and FPSIS (Zhipeng Zhao et al., 2019), QZS- inerter.
In this study, we combine a quasi-zero stiffness and frictional pendulum system (QZS-FPS). Looking at the literature, we have neither seen the work of another identical research team, nor obtained the same results.
In fact, many consider the negative stiffness isolator to be the QZS, but for the purposes of this work, the QZS used is an isolator that combines positive and negative stiffness to achieve near-zero stiffness. Combined with the FPS, the aim is to create a system with low stiffness around the equilibrium point, effectively isolating the more destructive low-frequency vibrations and dissipating energy by limiting residual displacements.
In relation to the work [9, 17, 29]
[9] develops a simplified and accurate model of seismic isolators for cost-effective protection of structures in seismic zones in developing countries. This work uses the Unbonded Fiber Reinforced Elastomeric Isolator, which does not necessarily combine a QZS and an FPS.
[17] studies the effectiveness of the Nonlinear Friction Pendulum in the stability of trains and railway tracks during earthquakes. This system cannot easily be confused with the QZS-FPS system. The disadvantage of the non-linear friction pendulum is that it increases lateral wheel-rail misalignments at the ends of the beam, causing an increase in the lateral wheel-rail force that can lead to uplift. The system proposed by [17] requires further study (the combined impact with other devices (e.g. rail dampers, anti-drift systems) has not been explored, which could alter the overall effectiveness). A comparison of this solution with other seismic isolation methods would also be relevant.
[29] Evaluates, via numerical simulations, the effectiveness of nonlinear isolators (springs and dampers) in reducing the oscillations of unbalanced rotating machines and minimizing the force transmitted to the ground over the entire operating range. [29] explores several types of nonlinear systems, but the case of QZS-FPS was not considered. Furthermore, [29] neglects the random behavior of nonlinear systems with stochastic signals.
This study builds on previous work carried out on QZS and FPS individually, and combines an analytical and numerical study of the QZS-FPS combination. The result is an isolator that performs better than any of the individual components.
In short, although the QZS (Quasi-Zero stiffness) and FPS (friction pendulum system) systems exist individually, this work stands out for: an in-depth analysis of the dynamic interactions between these two mechanisms, which had not been systematically explored in previous studies (eg. [9,17,29] ).the results presented in Figures 7 and 10 show a clear decrease in the amplitude of oscillations when FPS-QZS is coupled, compared with Figures 6 and 9, which represent the amplitude of oscillations in the case of a single QZS isolation and in the case of a single FPS isolation.
Comments 2:
Inadequate Comparative Analysis: Although a 30% displacement reduction is reported, the study fails to benchmark the QZS-FPS system against other hybrid configurations (e.g., QZS-TMD or FPS-negative stiffness systems). Consequently, the claimed superiority remains unsubstantiated. Furthermore, the physical rationale behind the enhanced performance of "soft-stiffness QZS-FPS" remains unexplained, weakening the conclusiveness of findings.Suggestion: If the author can reconstruct the isolation mechanism (such as introducing nonlinear adaptive control), provide multi system comparative data, or explore the unique applicability of QZS-FPS in new structures (such as super high-rise buildings), it may significantly enhance innovation.
Response 2:
Thank you for your comment,
A comparative analysis was carried out with the Negative Stiffness Integrated Tuned Mass Damper (NS-TMD) hybrid system studied by [36] and it was found that the results obtained by QZS-FPS are better than those presented by [36] in terms of damping the amplitudes of structure oscillations. Table 1 summarizing these differences has been added to the manuscript. However, we were unable to find any systems in the literature relating to QZS-TMD, but we'll keep looking.
Systems |
Improvements |
References |
NS-TMD |
Improve the stability of a bridge previously equipped with FPS. Reduces vibration with a maximum output amplitude of 0.081 m |
[25]
|
FPSIS |
Reduces vibrations with a maximum amplitude of 0.8 m. |
[27]
|
NS-FPS |
Develops an innovative solution to improve the seismic resilience of above-ground buildings and underground infrastructures (ASUS) by coupling negative stiffness and friction pendulum. |
[26]
|
FPS-TMD |
Reduces vibration with a maximum amplitude of 0.1 m. |
[27] |
QZS-FPS |
Reduces vibrations with a maximum amplitude of around 0.003 m for rigid ground and 0.007 m for soft ground. |
|

Reviewer 2 Report
Comments and Suggestions for Authors
This revised version submitted is not the last version, because several points of the review should be addressed. Please, check the revised version submitted.
Comments on the Quality of English Language13) A revision of the text is suggested to improve the quality of English language.
Author Response
This revised version submitted is not the last version, because several points of the review should be addressed. Please, check the revised version submitted.
Comments 1:
13) A revision of the text is suggested to improve the quality of English language.
Response 1:
Thank you for your comment. The entire text has been checked and all spelling and grammatical errors corrected.

Reviewer 3 Report
Comments and Suggestions for Authors
Authors should pay more attention and provide more information what they have done in the paper and what revision were made in revised manuscript (for example, the can higlight the place where they have made changes) when answering the reviewer.
Additionally, Comment 1, Comment 7 and Comment 8 of the reviewer is not made/satisfied although authors mention they reviewed. Therefore, I suspect the authors (a) do not actually read the comments, (b) they have not overlooked the reviewer suggestions for their study. In either cases the reply is not satisfactory.
Author Response
Authors should pay more attention and provide more information what they have done in the paper and what revision were made in revised manuscript (for example, the can higlight the place where they have made changes) when answering the reviewer.
Comments:
Additionally, Comment 1, Comment 7 and Comment 8 of the reviewer is not made/satisfied although authors mention they reviewed. Therefore, I suspect the authors (a) do not actually read the comments, (b) they have not overlooked the reviewer suggestions for their study. In either cases the reply is not satisfactory.
Comments 1:
Introduction section is weak and needs more improving. Authors should pay more attention to provide state of art review.
Response 1:
Thank you for your comment. The introduction has been improved and a table added and the improvements can be seen on lines 59 to 62.
Comments 7:
Discussion section is too superficial. Authors are suggested to improve discussion section including numerical results.
Response 7:
Thank you for your comment. Your comment has been taken into account and the discussion intensified. The improvements made can be appreciated on lines 354, 358-365, 366, 371-375.
Comments 8:
The conclusion section must also describe the limitations of the study and this section should be revised.
Response 8:
Thank you for your comment. We have indeed revised the conclusion and for this purpose, added the results obtained both with the harmonic and stochastic signal, then as far as possible, illustrate some limits of the work. The improvements made can be appreciated on lines 378-391.

Round 3
Reviewer 1 Report
Comments and Suggestions for Authors
Accept in present form.
Reviewer 2 Report
Comments and Suggestions for Authors
The revised manuscript can be recommended for publication.
Reviewer 3 Report
Comments and Suggestions for Authors
Authors have shown the changes made and provided more detailed comments for 1-3 and 7 and made required changes. Limitations is described and possible future studies are stated in the conclusion section.